

# Reliability of five-minute *vs.* one-hour heart rate variability metrics in individuals with spinal cord injury

Siriwipa Srirubkhwa[1], Lars Brockmann[2], Ratana Vichiansiri[1], Kenneth J. Hunt[2] and Jittima Saengsuwan[1,2]

[1] Department of Rehabilitation Medicine, Faculty of Medicine, Khon Kaen University, Khon Kaen, Thailand
[2] The Laboratory for Rehabilitation Engineering, Institute for Human Centred Engineering, Bern University of Applied Sciences, Biel, Switzerland

Corresponding author
Jittima Saengsuwan,
sjittima@kku.ac.th

## ABSTRACT

**Background:** A previous study showed low reliability of 1-h HRV outcomes in participants with spinal cord injury (SCI), but it was not certain whether the low reliability was due to the unrestricted activity of participants. We aimed to investigate test-retest reliability of HRV metrics in individuals with SCI using a 1-h measurement in a supine position.

**Methods:** Individuals with SCI underwent two sessions of 1-h recording of the time between consecutive R waves (RR-intervals) in a supine position. HRV outcomes were obtained from a single 5-min data segment and for the full 1-h recording. HRV parameters of interest were: standard deviation of all normal-to-normal R-R intervals (SDNN) and square root of the mean of the squared differences between successive R-R intervals (RMSSD) (time domain); and high frequency power (HF), low frequency power (LF), very low frequency power (VLF), ultra-low frequency power (ULF) and total power (TP) (frequency domain). Relative reliability was assessed by intraclass correlation coefficient (ICC). Absolute reliability was assessed by coefficient of variation (CV) and Bland-Altman limits of agreement (LoA).

**Results:** Data from 37 individuals (14 with tetraplegia and 23 with paraplegia) were included. Relative reliability was higher for the 1-h (ICCs ranged from 0.13–0.71) than for the 5-min duration (ICCs ranged from 0.06–0.50) in the overall SCI group for all HRV metrics. Participants with tetraplegia had lower relative reliability compared to participants with paraplegia in all HRV metrics for the 5-min duration (ICCs ranged from −0.01–0.34 *vs.* 0.21–0.57). For the 1-h duration, participants with paraplegia showed higher relative reliability than participants with tetraplegia in all HRV metrics (ICCs ranged from 0.18–0.79 *vs.* 0.07–0.54) except TP (ICC 0.69 *vs.* 0.82). In terms of absolute reliability, the CVs and LoAs for the 1-h duration were better than for the 5-min duration. In general, time domain metrics showed better reliability than frequency domain metrics for both durations in participants with tetraplegia and paraplegia. The lowest CV and narrowest 95% LoA were found for SDNN in 5-min and 1-h durations overall and in both lesion levels.

**Conclusions:** The supine position did not provide better reliability compared to unrestricted activity in participants with SCI. HRV analysis using a 5-min duration is of limited value in SCI due to poor reliability. For the 1-h analysis duration,

interpretation of the reliability of HRV varies according to lesion level: it is recommended to take lesion level into account when interpreting reliability measures.

## INTRODUCTION

Heart rate variability (HRV) is the physiological phenomenon of variation in the time interval between consecutive heartbeats which is generated by the SA node in the right atrium of the heart (*Shaffer & Ginsberg, 2017*). Variations in heart rate are analyzed by different methods such as time domain, frequency domain or combined time-frequency measures (*Shaffer & Ginsberg, 2017*). HRV is usually measured with short (approximately 5 min) and long (24-h) durations (*Malik et al., 1996*; *Shaffer & Ginsberg, 2017*). Interestingly, short term and long term HRV reflect different underlying physiological processes. Long term HRV is attributed to changes in circadian rhythm, core body temperature, the renin-angiotensin system, and the sleep cycle (*Malik et al., 1996*; *Shaffer, Meehan & Zerr, 2020*). Short term HRV is thought to be generated by four sources: (1) interactions between sympathetic and parasympathetic autonomic regulation, (2) respiration, (3) baroreceptor reflex, that regulates blood pressure, and (4) rhythmic adjustment in blood vessel diameter (*Shaffer & Ginsberg, 2017*). Thus, long term HRV is not interchangeable with short term HRV (*Shaffer, Meehan & Zerr, 2020*). Long term HRV has been found to be a powerful and independent predictor of an adverse event such as morbidity or death in multiple patient populations (*Fang, Wu & Tsai, 2020*; *Hohnloser et al., 1997*; *Nunan, Sandercock & Brodie, 2010*; *Patel et al., 2017*). However, some studies demonstrated that short term or even ultra-short-term HRV recording (<5 min) might also be used to predict adverse events (*Fei et al., 1996*; *Karp et al., 2009*; *Kautzner et al., 1998*; *La Rovere et al., 2003*).

To employ HRV in clinical practice, apart from its validity, the HRV metrics should be reliable and therefore applicable across individual patients. For reliability analysis, it is necessary to estimate outcome variability within individuals and in different populations. It was found that clinical populations showed poorer reliability than healthy subjects (*La Fountaine et al., 2010*; *Lord et al., 2001*). Additionally, reliability was poorer during interventions such as tilt or pharmacological stimulation compared to rest (*Sandercock, Bromley & Brodie, 2005*). *Sandercock, Bromley & Brodie (2005)* stated that describing HRV in general as a reliable measurement technique is an oversimplification, because the results of reliability studies are heterogenous and depend on a number of factors, thus studies in different clinical populations are required to assess HRV reliability.

The spinal cord is a compact bundle of neural structures that facilitate the transmission of both motor and sensory information between the brain and the rest of the body. Consequently, any damage to the spinal cord can disrupt the conduction of signals associated with sensory, motor, and autonomic functions across the site of the injury (*Rupp et al., 2021*). Spinal cord injury (SCI) can be categorized into two primary groups,

determined by the level of the lesion: tetraplegia refers to the impairment or loss of motor and/or sensory function in the cervical segments of the spinal cord, while paraplegia refers to the impairment or loss of motor and/or sensory function in the thoracic, lumbar or sacral segments (*Rupp et al., 2021*). Individuals with SCI also suffer from consequences of autonomic dysfunction, including issues such as sexual dysfunction as well as bowel and bladder problems (*Hou & Rabchevsky, 2014*). Regarding the autonomic control of the heart, the parasympathetic system is undamaged in individuals with SCI, since its regulation is from the vagal nuclei in the brain stem (*Takahashi et al., 2007*). On the other hand, sympathetic control of the heart relies on bulbospinal input *via* the cervical region and spinal sympathetic preganglionic neurons located in the upper half (T1–T6) of the thoracic segments. As a result, those with injuries at or above T6 have compromised sympathetic control of the heart (*Fossey et al., 2022*; *Malmqvist et al., 2015*; *Rodrigues et al., 2016*). Research indicates that individuals with SCI exhibit lesion-dependent impairment in resting cardiovascular function. Among these individuals, those with the highest injuries (tetraplegia) demonstrate the greatest degree of cardiovascular dysfunction compared to individuals with high thoracic (T1–T6) and low thoracic (below T6) paraplegia (*West, Mills & Krassioukov, 2012*). Notably, cardiovascular disease is the leading cause of death after SCI (*Sabre et al., 2013*; *Savic et al., 2017*). Additionally, individuals with SCI, particularly individuals with tetraplegia, have higher risk of cardiovascular events and deaths when compared to their able-bodied counterparts (*Chamberlain et al., 2019*; *Cragg et al., 2013*; *Jia et al., 2023*). Apart from an altered metabolism and decrease in physical activity, autonomic dysfunction is one of the proposed contributing factors (*Raguindin et al., 2021*). In comparison with healthy counterparts, participants with SCI had lower HRV values for both time and frequency domain metrics (*Rodrigues et al., 2016*; *Serra-Ano et al., 2015*).

Our previous study in 45 individuals with SCI and without any restriction on activity showed that when the duration of HRV recording was shorter (*i.e.* 1, 3 or 6 h), the reliability was lower than long-duration recording (24 h) (*Ruangsuphaphichat et al., 2023*). However, 24-h recording may not be convenient in clinical practice and it has not been integrated in general medical care (*Shaffer, Meehan & Zerr, 2020*). Additionally, the low reliability observed for 1-h recording in our previous study may also have resulted from placing no restriction on activity. We thus aimed to investigate test-retest reliability of short term HRV measurement, namely 5-min and 1-h durations, when participants rest in the supine position. The 5-min duration was chosen because it is a standard and generally used method that requires a short time to perform (*Malik et al., 1996*; *Sassi et al., 2015*; *Shaffer & Ginsberg, 2017*). The 1-h duration was used in order to be able to establish the test-retest reliability of the ultra-low-frequency (ULF) outcome since the shortest time period for ULF to be valid is 1 h (*Tan & Jiang, 2013*).

## MATERIALS AND METHODS

### Participants

We studied individuals with SCI who were admitted at Srinagarind Hospital, which is a university hospital in the northeast region of Thailand, from September 2021 to January

2023. Inclusion criteria were SCI of any cause, and age ≥18 years. Exclusion criteria were abnormal breathing pattern (respiratory rate >20 breaths/min or <10 breaths/min), concomitant cardiac or neurological disease and fever (body temperature ≥37.8 deg C). Ethical approval for this study was obtained from the Khon Kaen University Committee for Ethics in Human Research (ref. HE641355). Written informed consent was obtained from all participants before inclusion.

## HRV measurement protocol

Participants underwent two test sessions. Each session was separated by at least 24 h. The recording time started at the same time for both sessions, *i.e.*, between 9 am and 10 am. Prior to each test, individuals were instructed to refrain from smoking, and from drinking caffeine or alcohol for 24 h before the measurement. During each test session, participants lay in a supine position for 1 h while they wore a chest belt sensor (Polar H10; Polar Electro Oy, Kempele, Finland) and a wristwatch receiver (Polar V800) to record the raw RR intervals. Participants were instructed to breathe normally during recordings.

## Outcomes and data processing

Raw RR intervals were exported as a text file to HRV analysis software implemented in Matlab (The Mathworks, Inc., Natick, MA, USA). Some recordings were invalid because of poor signal quality. The remaining data sets were preprocessed for artefact detection and removal. Artefact detection was performed using two methods: (i) maximal and minimal values for plausible RR values were defined (min = 400 ms; max = 1,650 ms), (ii) the difference between two successive RR intervals was set to be at a maximum of ±20% of the previous value. For the removal of the detected artefacts, special care was taken not to add spurious information to the original data sets by removing any artificially introduced combinations of two successive RR intervals from the analysis. The recording was done for 1-h, and the 5-min segment to be used for short-term analysis was obtained from minutes 5 to 10 of the recording.

For the time-domain analysis, the metrics used were standard deviation of all normal-to-normal R-R intervals (SDNN) and square root of the mean of the squared differences between successive R-R intervals (RMSSD). In the frequency-domain analysis, power in the high frequency (HF, 0.15–0.4 Hz), low frequency (LF, 0.04–0.15 Hz), very low frequency (VLF, 0.0033–0.04 Hz), and ultra-low frequency (ULF, <0.0033 Hz) bands, together with total power (TP) was calculated. The Lomb-Scargle least squares spectral analysis method was used because it is more appropriate for spectral analysis when the data are irregularly sampled such as in RR time series (*Fonseca et al., 2013*; *Krafty et al., 2014*; *Stewart et al., 2020*).

## Sample size calculation

Sample size was calculated based on the sample size formula for ICC (*Bujang & Baharum, 2017*). Using a prespecified reliability of 0.70 and expected reliability of 0.85, a power of 90% and a significance level of 0.05, the sample size required was 56 participants.

We added subjects to cover an expected dropout rate of 15%. Thus, we finally required 66 participants.

## Statistical analysis

The outcomes are presented as median (with 25th and 75th percentiles) because the data were found to be not normally distributed. Extreme outliers were identified and excluded from data analysis when the changes in HRV metric values were more than three interquartile ranges (IQR) above Q3 or below Q1 (*Jones, 2019*). Wilcoxon signed-rank tests were used to test paired differences for each participant with significance level set to $\alpha = 0.05$. Relative test-retest reliability was analysed using the intraclass correlation coefficient $ICC_{3,1}$ and is presented as ICC and 95% confidence interval (CI). $ICC \geq 0.75$ represents excellent reliability, $ICC < 0.4$ is poor reliability and ICC between these ranges is regarded as moderate to good reliability (*Andresen, 2000*). Absolute reliability was evaluated using the coefficient of variation (CV) (*Bland & Altman, 1996*) and Bland-Altman limits of agreement (LoA) (*Bland & Altman, 1986*). Regarding Bland-Altman analysis, since all data were heteroscedastic, the data were log-transformed prior to analysis. Then the data are transformed back and the results are presented in the Bland-Altman plots as a linear function $\pm b\bar{x}$ (*Euser, Dekker & le Cessie, 2008*), where $\bar{x}$ is the data mean and $b$ is the slope of the LoA. The statistical analyses were performed using SPSS (IBM SPSS Statistics for Windows, Version 28.0. Armonk, NY: IBM Corp).

## RESULTS

### Demographic data

Sixty-six individuals with SCI were recruited in accordance with the inclusion criteria listed above (section Participants). During HRV data processing, some data sets were invalid. The exclusion of data from 29 individuals was due to signal gap (58.9%), noisy signal (30.4%) and multiple skipped heart rate measurements (16.1%). As a result, data from 37 individuals were included for analysis (14 tetraplegia and 23 paraplegia). More than half of the participants were male (54%). The mean age was 49.5 years and the median duration after SCI was 7.6 years (Table 1).

Regarding 5-min HRV, 7 HF values and 1 LF value could not be obtained from the analysis. Additionally, the removal of outliers led to an exclusion of 1 SDNN, 4 HF, 3 LF and 1 TP values. Since TP represents the combination of all frequency spectra, the exclusion of the above-mentioned frequency band data resulted in 25 data pairs for TP analysis. Regarding 1-h HRV, 5 HF values and 1 LF value could not be obtained from the analysis. The identification of outliers led to the exclusion of an additional 4 HF, 3 LF, and 1 VLF values, resulting in 27 data pairs for TP analysis.

### Overall reliability of HRV in SCI

There were no significant differences ($p > 0.05$) in any pairs of HRV values for both durations (5 min and 1 h) in the overall analysis (*i.e.*, without consideration of lesion level). For the 5-min duration overall, HF had moderate to good relative reliability (ICC = 0.50), but all other outcomes had poor reliability (ICC 0.06–0.36). For the 1-h duration, overall

**Table 1 Demographic data (n = 37).**

| Variables | n (%) |
|---|---|
| **Sex** | |
| Male | 20 (54.1) |
| Female | 17 (45.9) |
| **Age** (years), Mean (SD) | 49.5 (13.8) |
| Range (years) | 26–76 |
| **SCI level** | |
| *Tetraplegia:* | |
| Complete | 4 (10.8) |
| Incomplete | 10 (27.0) |
| *Paraplegia:* | |
| Complete | 6 (16.2) |
| Incomplete | 17 (45.9) |
| **Cause of SCI** | |
| *Traumatic* | 20 (54.1) |
| *Non-Traumatic* | |
| Degenerative | 3 (8.1) |
| Inflammatory | 1 (2.7) |
| Neoplastic | 8 (21.6) |
| Infection | 3 (8.1) |
| Other | 2 (5.4) |
| **Duration of SCI** (years), Median (p25, p75) | 7.6 (26, 146.5) |
| Range (years) | 0.3–19.2 |
| **Underlying diseases** | |
| No underlying disease | 28 (75.7) |
| Hypertension | 4 (10.8) |
| Dyslipidemia | 3 (8.1) |
| Other | 5 (13.5) |
| **Medications** | |
| **Antihypertensive** | 4 (10.8) |
| Amlodipine | 3 (8.1) |
| Enalapril | 1 (2.7) |
| Alpha-blockers | 7 (18.9) |
| **Anticholinergics** | 27 (73.0) |
| Oxybutynin | 23 (62.2) |
| Trospium | 7 (18.9) |
| Detrusitol | 1 (2.7) |
| **Medications for neuropathic pain** | 24 (64.9) |
| Gabapentin | 20 (54.1) |
| Pregabalin | 3 (8.1) |
| Amitriptyline | 4 (10.8) |
| **Antispastic** | 24 (64.9) |
| Baclofen | 20 (54.1) |
| Tizanidine | 3 (8.1) |
| Clonazepam | 11 (29.7) |
| Diazepam | 1 (2.7) |

**Note:**
p25, 25[th] percentile; p75, 75[th] percentile; SD standard deviation.

**Table 2  Test-retest reliability; 5-min and 1-h recordings; $n = 37$.**

| | Day 1, Median (p25, p75) | Day 2, Median (p25, p75) | p-value | ICC (95% CI) | CV | 95% LoA |
|---|---|---|---|---|---|---|
| **5-min** | | | | | | |
| RR interval (ms) | 840.0 (138.0) | 815.0 (111.0) | | | | |
| HR (bpm) | 73.4 (12.6) | 75.0 (10.5) | | | | |
| SDNN (ms), ($n = 36$) | 35.9 (22.2, 49.5) | 38.1 (28.3, 46.5) | 0.36 | 0.27 [−0.05 to 0.54] | 47.2 | ±0.95 $\bar{X}$ |
| RMSSD (ms) | 19.4 (12.1, 26.1) | 21.0 (14.7, 33.2) | 0.44 | 0.36 [0.04–0.61] | 67.3 | ±1.22 $\bar{X}$ |
| HF (ms$^2$), ($n = 27$) | 151.7 (31.9, 417.0) | 224.5 (88.3, 542.8) | 0.36 | 0.50 [0.15–0.74] | 158.0 | ±1.71 $\bar{X}$ |
| LF (ms$^2$), ($n = 33$) | 259.1 (122.2, 665.9) | 298.8 (172.0, 556.0) | 0.90 | 0.06 [−0.27 to 0.39] | 151.6 | ±1.68 $\bar{X}$ |
| TP (ms$^2$), ($n = 25$) | 922.2 (493.2, 2,304.3) | 1,434.1 (761.0, 2,011.5) | 0.34 | 0.35 [−0.04 to 0.65] | 86.6 | ±1.33 $\bar{X}$ |
| **1-h** | | | | | | |
| RR interval (ms) | 854.0 (134.0) | 843.0 (109.0) | | | | |
| HR (bpm) | 72.1 (12.1) | 72.4 (9.9) | | | | |
| SDNN (ms) | 57.5 (47.8, 71.1) | 61.2 (48.4, 79.4) | 0.14 | 0.61 [0.37–0.78] | 28.8 | ±0.65 $\bar{X}$ |
| RMSSD (ms) | 22.9 (14.0, 29.7) | 25.2 (16.7, 33.6) | 0.46 | 0.46 [0.15–0.68] | 51.4 | ±1.03 $\bar{X}$ |
| HF (ms$^2$), ($n = 28$) | 270.0 (153.1, 867.5) | 432.7 (166.2, 646.2) | 0.35 | 0.66 [0.40–0.83] | 75.9 | ±1.30 $\bar{X}$ |
| LF (ms$^2$), ($n = 33$) | 467.1 (182.5, 859.1) | 615.0 (261.2, 880.6) | 0.55 | 0.13 [−0.21 to 0.45] | 97.4 | ±1.45 $\bar{X}$ |
| VLF (ms$^2$), ($n = 36$) | 976.4 (479.5, 1,880.1) | 1,234.3 (728.9, 2,053.7) | 0.31 | 0.28 [−0.04 to 0.55] | 67.5 | ±1.18 $\bar{X}$ |
| ULF (ms$^2$) | 1,297.0 (705.1, 2,620.8) | 1,593.6 (875.3, 3,810.2) | 0.22 | 0.40 [0.10–0.64] | 103.2 | ±1.47 $\bar{X}$ |
| TP (ms$^2$), ($n = 27$) | 3,198.8 (1,819.6, 6,140.9) | 3,622.6 (1,687.0, 7,245.0) | 0.70 | 0.71 [0.46–0.86] | 47.5 | ±0.97 $\bar{X}$ |

**Note:**
CI, confidence interval; CV, coefficient of variation; HF, high frequency power; ICC, intraclass correlation coefficient; LF, low frequency power; LoA, limits of agreement; p25, 25th percentile; p75, 75th percentile; RMSSD, root mean square of successive differences between normal heartbeats; SDNN, standard deviation of all normal-to-normal R-R intervals; TP, total power; ULF, ultra-low frequency power; VLF, very low frequency power; $\bar{X}$ data mean.

relative reliability was moderate to good for SDNN, RMSSD, HF, ULF and TP (ICC 0.40–0.71), while it was poor for LF and VLF (ICCs of 0.13 and 0.28, respectively). For all outcomes, relative reliability was higher (higher ICC) for the 1-h than for the 5-min duration.

In terms of absolute reliability, the CVs for the 1-h duration were better than for the 5-min duration. In general, time domain metrics showed better reliability than frequency domain metrics for both durations. The CVs for the 5-min HRV outcomes were 47.2% (SDNN) and 67.3% (RMSSD) for the time domain metrics and in the range of 86.6% to 158.0% for the frequency domain metrics. For the 1-h HRV outcomes, CVs were 28.8% (SDNN) and 51.4% (RMSSD) for the time domain metrics and in the range of 47.5% to 103.2% for the frequency domain metrics. The lowest CV and narrowest 95% LoA were found for SDNN. Similar to the CVs, the limits of agreement were narrower for the 1-h duration compared to the 5-min duration for all outcomes (Table 2, Fig 1).

### Reliability of HRV classified by lesion level

The 5-min HRV outcomes showed a significant difference ($p = 0.034$) in HF within the group with paraplegia. However, there were no significant differences ($p > 0.05$) in any pairs of HRV values in the 5-min HRV outcomes of participants with tetraplegia. Similarly, no significant differences were found for the 1-h HRV outcomes in both groups.

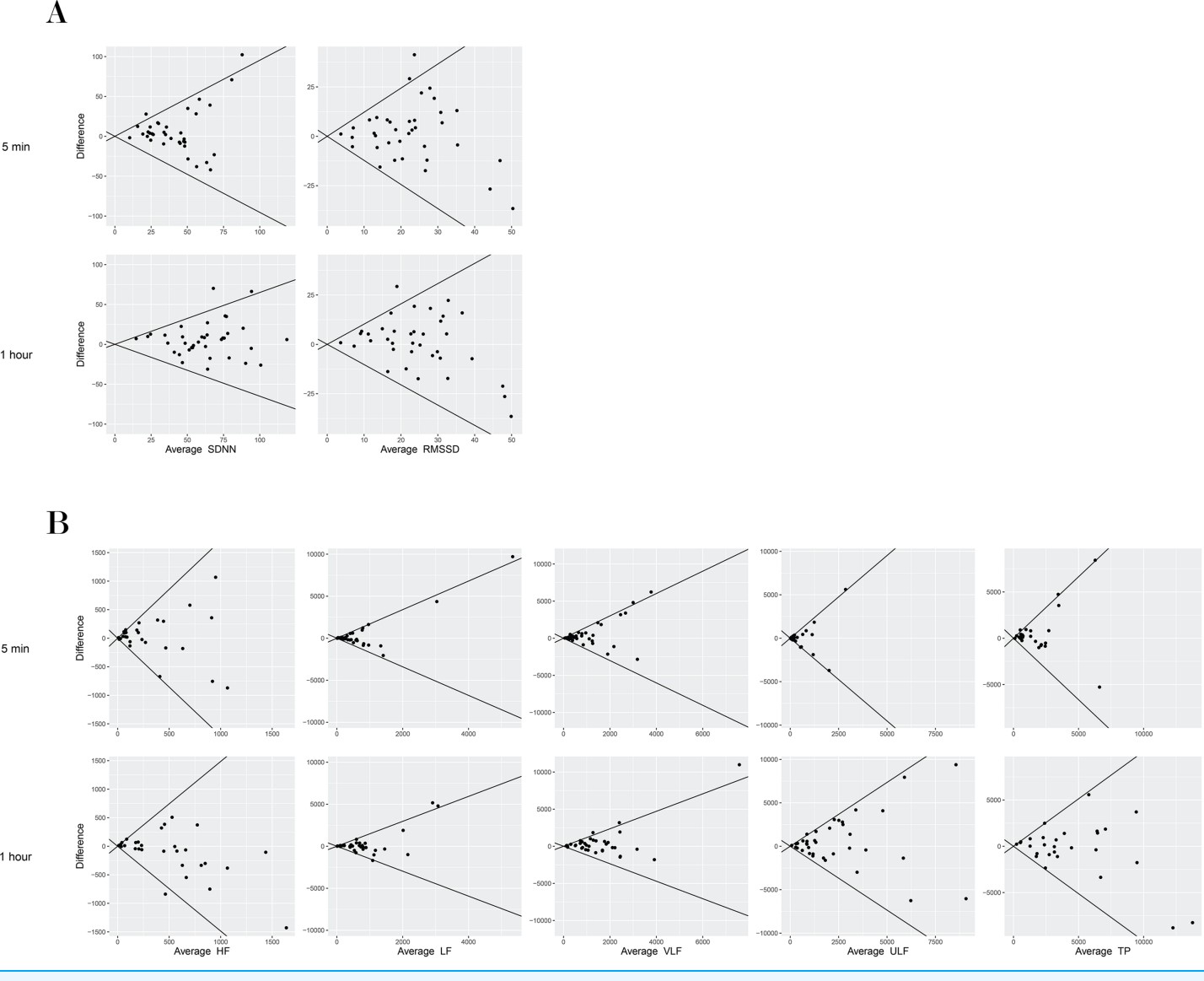

**Figure 1** **Bland–Altman plot of mean differences and 95% limits of agreement (LoA) among time domain HRV measures (5-min and 1-h) in a supine position in individuals with SCI.**

Regarding relative reliability, participants with tetraplegia generally had lower reliability compared to participants with paraplegia across most HRV metrics, both in 5-min and 1-h durations. Among all HRV metrics, LF exhibited the lowest relative reliability within both the 5-min and 1-h durations in both groups. Similar to what was observed in overall SCI, relative and absolute reliability improved with longer duration (1-h). For relative reliability, the three most reliable metrics were HF, RMSSD and TP for the 5-min duration and TP, HF and SDNN for the 1-h duration.

For the 5-min HRV, the CVs of participants with tetraplegia were poorer than participants with paraplegia in the frequency-domain metrics but not in the time-domain metrics. Regarding the 1-h HRV, the CVs of participants with tetraplegia were smaller

**Table 3 Test-retest reliability of 5-min and 1-h duration HRV classified by tetraplegia and paraplegia; n = 37.**

| | Day 1, Median (p25, p75) | Day 2, Median (p25, p75) | p-value | ICC (95% CI) | CV | 95% LoA |
|---|---|---|---|---|---|---|
| **5-min, Tetraplegia ($n = 14$)** | | | | | | |
| SDNN (ms), ($n = 13$) | 42.0 (31.4, 50.7) | 42.0 (36.0, 46.1) | 0.68 | 0.18 [−0.37 to 0.65] | 45.4 | ±0.94 $\bar{\text{X}}$ |
| RMSSD (ms) | 20.4 (14.6, 28.9) | 24.2 (17.9, 36.5) | 0.63 | 0.32 [−0.25 to 0.72] | 53.6 | ±1.07 $\bar{\text{X}}$ |
| HF (ms$^2$), ($n = 10$) | 359.4 (186.7, 721.9) | 307.8 (88.4, 542.7) | 0.49 | 0.34 [−0.35 to 0.78] | 184.6 | ±1.82 $\bar{\text{X}}$ |
| LF (ms$^2$), ($n = 12$) | 455.7 (226.9, 953.7) | 425.6 (150.3, 544.8) | 0.34 | −0.01 [−0.59 to 0.56] | 164.0 | ±1.76 $\bar{\text{X}}$ |
| TP (ms$^2$) ($n = 9$) | 1,711.6 (776.9, 2,487.9) | 1,794.1 (524.3, 2,204.2) | 0.91 | 0.21 [−0.44 to 0.74] | 77.3 | ±1.29 $\bar{\text{X}}$ |
| **5-min, Paraplegia ($n = 23$)** | | | | | | |
| SDNN (ms) | 32.9 (22.1, 48.2) | 38.1 (27.6, 46.9) | 0.44 | 0.34 [−0.08 to 0.66] | 48.2 | ±0.98 $\bar{\text{X}}$ |
| RMSSD (ms) | 18.5 (9.5, 24.7) | 20.4 (13.2, 32.1) | 0.73 | 0.38 [−0.04 to 0.68] | 75.1 | ±1.30 $\bar{\text{X}}$ |
| HF (ms$^2$), ($n = 17$) | 80.7 (31.0, 228.7) | 131.4 (90.2, 342.1) | 0.034 | 0.57 [0.16–0.82] | 142.3 | ±1.60 $\bar{\text{X}}$ |
| LF (ms$^2$), ($n = 21$) | 174.0 (80.5, 498.2) | 293.4 (191.7, 692.5) | 0.27 | 0.21 [−0.21 to 0.57] | 144.4 | ±1.62 $\bar{\text{X}}$ |
| TP (ms$^2$), ($n = 16$) | 637.5 (484.9, 2,085.0) | 1,200.2 (771.7, 1,719.6) | 0.32 | 0.50 [0.002–0.79] | 91.7 | ±1.38 $\bar{\text{X}}$ |
| **1-h, Tetraplegia ($n = 14$)** | | | | | | |
| SDNN (ms) | 58.5 (55.0, 71.1) | 65.3 (51.0, 84.7) | 0.079 | 0.54 [0.07–0.82] | 26.9 | ±0.58 $\bar{\text{X}}$ |
| RMSSD (ms) | 20.9 (14.0, 33.4) | 26.3 (18.9, 35.0) | 0.42 | 0.41 [−0.15 to 0.76] | 43.8 | ±0.92 $\bar{\text{X}}$ |
| HF (ms$^2$), ($n = 11$) | 617.5 (229.1, 978.7) | 461.0 (182.3, 699.4) | 0.067 | 0.54 [0.01–0.85] | 58.6 | ±1.15 $\bar{\text{X}}$ |
| LF (ms$^2$), ($n = 13$) | 681.5 (346.4, 859.1) | 6,19.0 (242.1, 816.6) | 0.89 | 0.07 [−0.53 to 0.59] | 92.5 | ±1.47 $\bar{\text{X}}$ |
| VLF (ms$^2$), ($n = 13$) | 1,321.7 (731.0, 1,820.0) | 1,363.7 (914.8, 2,089.9) | 0.38 | 0.19 [−0.34 to 0.64] | 81.5 | ±1.26 $\bar{\text{X}}$ |
| ULF (ms$^2$) | 1,201.6 (934.9, 2,620.8) | 1,775.4 (1,004.4, 3,971.7) | 0.091 | 0.20 [−0.27 to 0.62] | 98.3 | ±1.33 $\bar{\text{X}}$ |
| TP (ms$^2$), ($n = 10$) | 3,575.0 (1,819.6, 6,533.4) | 4,112.9 (2,752.4, 6,150.6) | 0.32 | 0.82 [0.44–0.95] | 41.5 | ±0.83 $\bar{\text{X}}$ |
| **1-h, Paraplegia ($n = 23$)** | | | | | | |
| SDNN (ms) | 55.7 (46.0, 71.6) | 57.0 (38.0, 79.4) | 0.71 | 0.66 [0.36–0.84] | 29.9 | ±0.69 $\bar{\text{X}}$ |
| RMSSD (ms) | 23.3 (9.3, 27.6) | 25.0 (15.2, 31.6) | 0.85 | 0.49 [0.10–0.75] | 55.7 | ±1.10 $\bar{\text{X}}$ |
| HF (ms$^2$), ($n = 17$) | 263.0 (143.2, 556.9) | 239.6 (150.0, 628.2) | 0.55 | 0.79 [0.51–0.92] | 86.3 | ±1.33 $\bar{\text{X}}$ |
| LF (ms$^2$), ($n = 20$) | 413.8 (173.0, 864.8) | 550.4 (271.9, 904.5) | 0.37 | 0.18 [−0.26 to 0.57] | 100.6 | ±1.41 $\bar{\text{X}}$ |
| VLF (ms$^2$) | 9,16.7 (448.5, 1,940.1) | 1,104.9 (655.1, 2,017.5) | 0.80 | 0.64 [0.31–0.83] | 59.2 | ±1.12 $\bar{\text{X}}$ |
| ULF (ms$^2$) | 1,297.0 (483.7, 2,665.8) | 1,555.1 (578.6, 3,810.1) | 0.82 | 0.48 [0.08–0.74] | 106.1 | ±1.53 $\bar{\text{X}}$ |
| TP (ms$^2$), ($n = 17$) | 3,163.3 (2,206.4, 5,612.1) | 3,117.4 (1,412.1, 7,245.0) | 0.68 | 0.69 [0.33–0.88] | 50.9 | ±1.05 $\bar{\text{X}}$ |

**Note:**
CI, confidence interval; CV, coefficient of variation; HF, high frequency power; ICC, intraclass correlation coefficient; LF, low frequency power; LoA, limits of agreement; p25, 25th percentile; p75, 75th percentile; RMSSD, root mean square of successive differences between normal heartbeats; SDNN, standard deviation of all normal-to-normal R-R intervals; TP, total power; ULF, ultra-low frequency power; VLF, very low frequency power; $\bar{\text{X}}$ data mean.

than participants with paraplegia except for VLF. The smallest CVs and narrowest LoAs were found for the time domain metrics (SDNN and RMSSD) in both groups (Table 3).

## DISCUSSION

We aimed to investigate test-retest reliability of short term HRV measurement, namely 5-min and 1-h durations, when participants rest in the supine position. In general, participants showed a lower degree of HRV for 5-min and 1-h metrics compared to healthy subjects. For 5-min HRV, the medians of HF, LF and TP were 151.7, 259.1 and 922.2 ms$^2$, respectively (Table 2). The corresponding mean HRV values in healthy subjects were reported elsewhere to be 975, 1,170 and 3,466 ms$^2$ (*Malik et al., 1996*). Regarding 1-h

HRV outcomes, mean SDNN and RMSSD were 90.4 and 47.8 ms in healthy subjects (*Evrengul et al., 2006*) and the medians of SDNN and RMSSD were 57.5 and 22.9 ms in our population (Table 2). The finding of lower HRV values in participants with SCI compared to healthy participants is consistent with multiple previous studies (*La Fountaine et al., 2010*; *Rodrigues et al., 2016*; *Serra-Ano et al., 2015*; *Thayer et al., 2016*).

SDNN showed lowest CV and narrowest 95% LoA. However, the overall CV of SDNN between test and retest measures of the 5-min HRV was large (47.2%) compared to previous reports in healthy participants (8.0%) (*Sinnreich et al., 1998*). Additionally, populations with SCI had lower relative reliability (ICC of 0.27 and 0.36 for SDNN and RMSSD, respectively) compared to healthy participants measured for the same duration and in the same position (ICC of 0.82 and 0.76 for SDNN and RMSSD) (*Pinna et al., 2007*).

Our study agrees with previous studies which reported that frequency domain measures (HF and LF) demonstrated larger variation and higher CV compared to time domain measures (*Nunan, Sandercock & Brodie, 2010*; *Pinna et al., 2007*; *Ponikowski et al., 1996*; *Salo et al., 1999*). The relative reliability of frequency domain measures was lower than in healthy participants. For example, for 5-min HRV reliability, our ICCs for HF and LF were 0.50 and 0.06 while the respective values in healthy participants were 0.63 and 0.79 (*Pinna et al., 2007*). Additionally, for the 1-h duration in a supine position, the ICCs of HF and LF were 0.66 and 0.13 in our study, and 0.81 and 0.86 in normal subjects (*da Cruz et al., 2019*). Our study shows much lower reliability of 1-h HRV in frequency domain measures in participants with tetraplegia (ICCs of HF and LF were 0.54 and 0.07) compared to a previous study: *La Fountaine et al. (2010)* found that the ICCs for HF and LF were 0.66 and 0.44 in individuals with tetraplegia and the ICCs were 0.90 and 0.74 in healthy individuals. To consider the least reliable HRV parameters among LF and HF, the results were inconsistent among studies: some studies found lower relative reliability of LF and some found the opposite (*Freed et al., 1994*; *Nunan, Sandercock & Brodie, 2010*; *Sinnreich et al., 1998*). Our study showed that LF had lowest relative reliability in participants with SCI based on 1-h HRV metrics.

Regarding absolute reliability, the trend of higher variability in the frequency domain was found in both healthy participants and in populations with diseases such as chronic heart failure. *Nunan, Sandercock & Brodie (2010)* showed that the overall CVs for SDNN and RMSSD were 32% and 37% while the CVs in LF and HF were 56% and 116% for the short term recording (5-min) of HRV in healthy adults. *Ponikowski et al. (1996)* demonstrated that the CVs of HF, LF, VLF and TP were 66.4%, 81.5%, 56.0% and 45.9% while the CV of SDNN was 25.4% in 5-min recording data in patients with chronic heart failure. *Lord et al. (2001)* showed that the CV for LF was 45% in healthy participants and 76% in cardiac transplant recipients. Our study demonstrated the CV of SDNN and RMSSD of 47.2% and 67.3% while the CV of HF, LF, VLF and TP were 158.0, 151.6, 114.8 and 86.6%. LF, HF, VLF and ULF seem to have lower absolute reliability based on CV and LoA compared to TP.

Poor reliability of short term (5-min) HRV has been identified in our study. This is in line with other studies which were done in patients with chronic heart failure, type-2 diabetes and cardiac transplant recipients (*Lord et al., 2001*; *Ponikowski et al., 1996*; *Sacre*

*et al., 2012*). One possible explanation is the low initial HRV values reported in clinical populations, therefore, slight changes in the test-retest value can result in a high CV. Lower absolute values among participants may also lead to lower ICC (*Sandercock, Bromley & Brodie, 2005*). Regarding 5-min and 1-h reliability of HRV, we found that the supine position did not provide better reliability compared to unrestricted activity in participants with SCI (*Ruangsuphaphichat et al., 2023*). This has to be interpreted with caution. Firstly, the results not showing improvement in reliability may be partially explained by differences in group proportions: the present study had a higher proportion of participants with tetraplegia (37.8%) compared to the previous study (22.2%) (*Ruangsuphaphichat et al., 2023*). We also had data from participants with complete tetraplegia while the former study did not. We found that individuals with tetraplegia showed poorer HRV reliability compared to those with paraplegia. This may be explained by the alteration in cardiogenic autonomic control in participants with higher lesion level, which results in higher inter-day variation of HRV (*Bauman et al., 2012*). Secondly, the influence of different body positions might contribute to the observed differences in reliability results. In a previous study, a reliability analysis of HRV in 37 non-SCI young men in different positions demonstrated noteworthy differences. The ICCs of HRV in the supine position were found to be lower than in the standing position. Specifically, the ICCs were as follows: log of SDNN (ICC of 0.60 *vs* 0.88 (supine *vs.* standing)), log of RMSSD (ICC of 0.74 *vs.* 0.86) and log of HF (ICC of 0.81 *vs.* 0.84). However, no such trend was observed for LF (ICC of 0.63 *vs.* 0.32) (*da Cruz et al., 2019*). This might imply that distinct body postures influence the reliability of HRV, particularly in the context of individuals with SCI. Given these findings, there emerges a potential need for further investigation into the impact of different postures on HRV metrics and their reliability within the SCI population.

Our participants had various medications, and some medications have been shown to alter HRV, *e.g.*, gabapentin, antihypertensive medication or tricyclic antidepressants (*Ermis et al., 2010*; *Miyabara et al., 2017*; *Pavithran et al., 2010*; *van Zyl, Hasegawa & Nagata, 2008*). However, since the medications were taken at regular times, these should not have directly affected test-retest differences. Our study had too few participants to conduct a robust comparison of HRV data in individuals with respect to whether there was a complete or incomplete lesion, which may lead to additional differences in reliability of HRV. Multiple invalid data sets due to poor signal recording has to be improved for clinical application, *e.g.*, by careful monitoring or regularly checking the data recording during the test period. Our results were limited to the supine position, and the single centre study may limit data generalization. It may be possible that different positions, or measurements with paced breathing, may lead to different reliability outcomes (*Nunan, Sandercock & Brodie, 2010*; *Pinna et al., 2007*; *Sinnreich et al., 1998*; *Tonello et al., 2015*). Additionally, the inclusion of age- and sex-matched able-bodied controls might have provided more comparable data in the same measurement setting and allowed more insightful interpretation. For future studies, emphasis on examining reliability in different body positions or during paced breathing sessions might offer avenues to improve the reliability of HRV measurements within this population. By including a larger number of

participants, it would enable the analysis of the potential impact of variables such as the severity and completeness of the lesion.

## CONCLUSIONS

The supine position did not provide better reliability compared to unrestricted activity in participants with SCI. HRV analysis using a 5-min duration is of limited value due to poor reliability in individuals with SCI. For the 1-h analysis duration, SDNN, RMSSD, HF and TP have moderate to excellent reliability overall and when grouped based on lesion level; ULF has moderate to good reliability overall and for paraplegia; VLF has moderate to good reliability for paraplegia only; but LF has poor reliability overall and in both groups. Interpretation of the reliability of HRV varies according to lesion level: it is recommended to take lesion level into account when interpreting reliability measures.

### Funding

The work was funded by a grant from the Swiss National Science Foundation as part of the project 'Heart Rate Variability, Dynamics and Control During Exercise' (Kenneth J. Hunt, Lars Brockmann, Jittima Saengsuwan; SNSF Grant Ref. 320030-185351). This study also received funding from the Faculty of Medicine, Khon Kaen University, Thailand (Siriwipa Srirubkhwa; Grant number IN65226). The funders had no role in study design, data collection and analysis, decision to publish, or preparation of the manuscript.

### Grant Disclosures

The following grant information was disclosed by the authors:
The work was funded by a grant from the Swiss National Science Foundation: 320030-185351.
Faculty of Medicine, Khon Kaen University, Thailand: IN65226.

### Competing Interests

The authors declare that they have no competing interests.

### Author Contributions

- Siriwipa Srirubkhwa conceived and designed the experiments, performed the experiments, analyzed the data, prepared figures and/or tables, authored or reviewed drafts of the article, and approved the final draft.
- Lars Brockmann conceived and designed the experiments, analyzed the data, prepared figures and/or tables, authored or reviewed drafts of the article, and approved the final draft.
- Ratana Vichiansiri conceived and designed the experiments, performed the experiments, analyzed the data, prepared figures and/or tables, authored or reviewed drafts of the article, and approved the final draft.

- Kenneth J Hunt conceived and designed the experiments, analyzed the data, prepared figures and/or tables, authored or reviewed drafts of the article, and approved the final draft.
- Jittima Saengsuwan conceived and designed the experiments, performed the experiments, analyzed the data, prepared figures and/or tables, authored or reviewed drafts of the article, and approved the final draft.

## Human Ethics

The following information was supplied relating to ethical approvals (*i.e.*, approving body and any reference numbers):

Ethical approval for this study was obtained from the Khon Kaen University Committee for Ethics in Human Research (ref. HE641355). Written informed consent was obtained from all participants before inclusion.

## Data Availability

The raw data is available in the Supplemental File.

## Supplemental Information

Supplemental information for this article can be found online at http://dx.doi.org/10.7717/peerj.16564#supplemental-information.

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
