# Peer review of "Reliability of five-minute vs. one-hour heart rate variability metrics in individuals with spinal cord injury"

_PeerJ, doi:10.7717/peerj.16564_

## Round 0.1 · original submission · Major Revisions

1. The article is overall well written.
2. Please add a proper control group as suggested by reviewer 2.

·

Basic reporting

The manuscript by Srirubkhwa et al. is well written and clear. The literature referenced in the manuscript is apt and sufficient to show the contribution of the study in the field of heart rate variability metrics. The data is presented clearly and the results are relevant to the hypothesis. The limitations of the study have been clearly specified; however, it would be beneficial to include any required future work.

Experimental design

The research is scientifically sound and within the scope of the journal. The importance of the study is clearly communicated by the authors. The investigation is rigorous and has high ethical standard. The methods section is detailed and clear.

Validity of the findings

The data shown is statistically sound and accurate. The conclusions answer all the research questions raised.

Additional comments

• Please include a background on tetraplegia and paraplegia and its impact on lesion size in the Introduction section.
• Please add a subtitle “Demographic data” in the very beginning of Results section.
• Please refer to table 2 in the Discussion section (Line 221).

Reviewer 2 ·

Basic reporting

Title: Reliability of five-minute vs. one-hour heart rate variability metrics in individuals with spinal cord injury

In this study, the authors aimed to evaluate the reliability of 1-hr Heart Rate Variability (HRV) in patients with spinal cord injury (SCI) in a supine position, instead of allowing unrestricted activity to ascertain the effect of physical activity on the outcomes of HRV.

Results in the current study are interesting and of general interest. Current draft of the manuscript requires major revisions. Additional experiments including non-SCI individuals will make the study stronger and provide a better interpretation of the results obtained in the 1-hour studies.

Experimental design

1. How do the authors account for discrepancies in the results from previous 1hr study and the current study? Explanation provided in the discussion section is handwavy at the best. Trivial explanations such as technical differences/ equipment calibration etc. need to be ruled out and that is why having a control group, such as non-SCI individuals is strongly recommended to interpret the observed results.

2. What are the primary conclusions from this study? The conclusions regarding 1-hr results in this manuscript are contradictory to the authors’ previous study without proper explanation provided.

Validity of the findings

1. In the current draft, results read like observations. Please include interpretations of obtained results and describe them in more detail.

Additional comments

Title: Reliability of five-minute vs. one-hour heart rate variability metrics in individuals with spinal cord injury

In this study, the authors aimed to evaluate the reliability of 1-hr Heart Rate Variability (HRV) in patients with spinal cord injury (SCI) in a supine position, instead of allowing unrestricted activity to ascertain the effect of physical activity on the outcomes of HRV.

Results in the current study are interesting and of general interest. Current draft of the manuscript requires major revisions. Additional experiments including non-SCI individuals will make the study stronger and provide a better interpretation of the results obtained in the 1-hour studies.

Please address following questions/comments below.

1. Please make sure that the manuscript meets the journal’s style requirements, including text alignments, figure labels and general proofreading. This is strongly recommended.
2. Please rewrite the background section to cater to a wider audience. Current language is very specific for the people who work in this area. For others, it’s hard to understand the goals and rationale of this study.
3. How do the authors account for discrepancies in the results from previous 1hr study and the current study? Explanation provided in the discussion section is handwavy at the best. Trivial explanations such as technical differences/ equipment calibration etc. need to be ruled out and that is why having a control group, such as non-SCI individuals is strongly recommended to interpret the observed results.
4. Line 166 – “insert table 1”. The manuscript needs thorough proofreading.
5. Line 192 – “insert table 2”. The manuscript needs thorough proofreading.
6. Line 194 – “insert figure 1”. The manuscript needs thorough proofreading.
7. Line 215 – “insert table 3”. The manuscript needs thorough proofreading.
8. In the current draft, results read like observations. Please include interpretations of obtained results and describe them in more detail.
9. Please provide higher resolution of figure 1. Currently, it’s hard to read labels and text in the figure.
10. Please rewrite figure legends and add more information that describes that’s being shown in each panel. Additionally, including a title will help the readers too. Explain the figure in more detail. Tables need more description too.
11. What are the primary conclusions from this study? The conclusions regarding 1-hr results in this manuscript are contradictory to the authors’ previous study without proper explanation provided.

Annotated reviews are not available for download in order to protect the identity of reviewers who chose to remain anonymous.

---

## Round 0.2 · Minor Revisions

Please address the comments from reviewer 2 and resubmit the manuscript.

·

Basic reporting

No comment.

Experimental design

No comment

Validity of the findings

No comment

Additional comments

After the revision, everything looks good.

Reviewer 2 ·

Basic reporting

Title: Reliability of five-minute vs. one-hour heart rate variability metrics in individuals with spinal cord injury

In this study, the authors aimed to evaluate the reliability of 1-hr Heart Rate Variability (HRV) in patients with spinal cord injury (SCI) in a supine position, instead of allowing unrestricted activity to ascertain the effect of physical activity on the outcomes of HRV.

I thank the authors for incorporating suggested edits. Majority of the comments and questions have been addressed satisfactorily. Please see comments below that still need to be addressed before the manuscript can be accepted for publication.


Please address following questions/comments below.

1. Current resolution of figures is still quite poor. If the authors printed out the figure pages, they would find that it’s hard to read labels and text in the figures. Please be thorough with your revisions.
2. Detailed figure legend for Figure 2 is entirely missing from both the word document and the PDF version of the manuscript. Please make sure to include it.
3. Page 6, under the “Demographic data” – please mention the criteria for including the sixty-six individuals.

Experimental design

Experiments have been satisfactorily designed.

Validity of the findings

Results are subtle, but the authors have done a good job in explaining the limitations of the current study.

Annotated reviews are not available for download in order to protect the identity of reviewers who chose to remain anonymous.

---

## Round 0.3 · Minor Revisions

Thank you for addressing the reviewer's comments and resubmitting the revised version with track changes. In the materials and method section, the authors are requested to explain the sample size for this study. Please describe whether any power calculation was done to decide the final sample size.

---

## Round 0.4 · accepted · Accept

Please provide high resolution figures for publication and make sure to add the detail in all figure legends.